# Insufficient Physical Activity Is a Global Marker of Severity in Alcohol Use Disorder: Results from a Cross-Sectional Study in 382 Treatment-Seeking Patients

**DOI:** 10.3390/nu14234958

**Published:** 2022-11-23

**Authors:** Julia de Ternay, Agathe Larrieu, Laura Sauvestre, Solène Montègue, Monique Guénin, Christophe Icard, Benjamin Rolland

**Affiliations:** 1Service Universitaire d’Addictologie de Lyon (SUAL), Hôpital Édouard Herriot, Hospices Civils de Lyon, 69003 Lyon, France; 2Service Universitaire d’Addictologie de Lyon (SUAL), CH Le Vinatier, 69678 Lyon, France; 3Centre de Recherche en Neurosciences de Lyon (Psychiatric Disorders, PSYR2), Université Claude Bernard Lyon 1, Inserm U1028, CNRS UMR 5292, 69100 Lyon, France

**Keywords:** alcohol use disorder, physical activity, alcohol addiction, physical rehabilitation

## Abstract

Improving physical activity (PA) in patients with alcohol use disorder (AUD) has recently emerged as an important component of the global treatment strategy to improve drinking outcomes and quality of life. However, this new approach should focus on AUD patients with insufficient baseline PA and requires this subgroup to be better characterized. In a population of 382 treatment-seeking AUD patients, PA was assessed using the International Physical Activity Questionnaire, and participants were divided into two groups: insufficient PA group and sufficient PA group. The severity of the AUD was assessed using the DSM-5 criteria, the Alcohol Use Disorder Identification Test, and the Severity of Alcohol Dependence Questionnaire. In logistic regression models, individuals with insufficient PA were more likely than other AUD individuals to present a higher Body Mass Index (*p* < 0.001), a higher number of AUD DSM-5 criteria (*p* < 0.05), more frequent opioid use (*p* < 0.05), higher scores at the Fagerström Test for Nicotine Dependence (*p* < 0.001), State-Trait Anxiety Inventory (*p* < 0.001), impulsivity scale (*p* < 0.05), Pittsburgh Sleep Quality Inventory (*p* < 0.05), and lower WHO Quality of Life (*p* < 0.001) scores. In AUD, an insufficient baseline PA is associated with several markers of severity, and physical exercise interventions should be part of a multimodal treatment program integrating the global impairments of patients.

## 1. Introduction

According to the last Global status report on alcohol and health from the World Health Organization, in 2016, 2.3 billion people were current alcohol drinkers worldwide, and 5.1% of the adults older than 15 years old suffered from an Alcohol Use Disorder (AUD) [1]. Harmful use of alcohol was responsible for three million deaths (representing 5.3% of all deaths) [1]. In France, 10% of adults were using alcohol daily in 2020, and alcohol was responsible for 41,000 annual deaths [2]. The French national prevalence of AUD is estimated to be 7.0% [1]. AUD is the official term used in the Diagnostic and Statistical Manual of Mental Disorders (DSM-5) [2] for alcohol addiction. Eleven criteria have been included in the diagnosis of AUD that all reflect a progressive loss of control upon alcohol use, through symptoms such as craving, tolerance, or inability to stop using alcohol despite repeated negative consequences [2]. Several markers of severity are commonly used to assess AUD. First, the number of DSM-5 criteria for AUD is considered as one of the main severity features, ranging from “mild” (i.e., two or three criteria met out of eleven), to “moderate” (i.e., four or five criteria), and “severe” (i.e., six criteria or more) [2]. The average number of drinks per day or week also reflects AUD severity [3]. More specific tools have been developed to measure AUD severity, such as the Severity of Alcohol Dependence Questionnaire (SADQ) [4]. To some extent, the Alcohol Use Disorder Identification Test (AUDIT) score can also be used to assess the severity of AUD, even if it has not been specifically developed for this purpose [5]. Other factors of AUD severity are the comorbid conditions, in particular, the associated psychiatric and addictive disorders, which impair the AUD prognosis. Last, functional impairments, such as associated sleep disorders [6], or poor quality of life [7], are considered significant markers of AUD severity and outcome.

The level of physical activity (PA) is a well-known factor in many positive health outcomes in the general population. Several meta-analyses have shown the benefits of a moderate-to-high level of PA on cardiovascular outcomes such as the decrease in the risk of incident coronary heart disease and stroke among men and women [8], or the decrease in cardiovascular disease mortality and cardiovascular disease incidence [9]. Similarly, a meta-analysis of 71 studies has shown an inverse non-linear dose response between the effects of PA and cancer mortality in the general population, where a minimum of 2.5 h a week of moderate-intensity activity was associated with a decrease in cancer mortality in men and women [10]. In contrast, both the amount and frequency of alcohol use are associated with an increased risk of cardiovascular diseases [11], liver diseases [12], and cancers [13].

The level of PA is increasingly investigated as a factor of a favorable outcome of AUD.

In patients with AUD, programs aiming to enhance PA have demonstrated that such enhancement was associated with a reduction in the subsequent amounts of drinking [14,15,16], and levels of craving [17,18], and with an increase in the overall scores of quality of life [19]. Consequently, it has been proposed to integrate programs based on PA into the treatment schemes of AUD [20,21,22]. However, few previous studies have investigated whether the initial level of PA of treatment-seeking patients with AUD could reflect a dimension of severity or outcome of the disease. Some studies showed that the level of PA was significantly correlated with the levels of global functioning and quality of life of patients [23,24], but they had small-size samples, i.e., less than 50 participants. To our knowledge, no previous study has investigated the heterogeneity of PA levels in patients with AUD, and its association with common markers of severity and outcomes, such as the number of DSM-5 criteria met or the occurrence of psychiatric or addictive comorbidities. This was the objective of our study, which was conducted on a large sample of treatment-seeking patients with AUD.

## 2. Materials and Methods

### 2.1. Participants

Included participants were adult (aged 18 years old or more) male and female outpatients of a French expert university addiction consultation center who had been diagnosed with AUD during their first medical consultation, using the DSM-5 criteria [2]. Participants not fulfilling one or more of those inclusion criteria were excluded. The recruitment period was between September 2017 and September 2021. After the first consultation, patients systematically underwent a structured assessment battery, based on the multiple questionnaires detailed below. The questionnaires were filled out by participants within the university addiction consultation center and their answers were coded and stored in a numeric database by one of the investigators. It generally took one hour for the participants to fully complete the questionnaires. The investigator in charge of collecting the data was supervising the participants while they were filling out the questionnaires, to help them by clarifying the meaning of the questions if it was needed.

All patients gave consent for the use of their health data in the present study, which was approved by the French Data Protection Commission (No. MR-004-2020-006).

### 2.2. Measurements

The sociodemographic and clinical characteristics collected were age, gender, marital status (partner/no partner), level of education (total years of education after high school diploma), professional status (active/not active), and body mass index (BMI). The diagnosis of AUD was established using the DSM-5 criteria for substance use disorder [2] as well as its severity: mild (2 to 3 criteria), moderate (4 to 5 criteria), and severe (6 or more criteria).

The level of PA was assessed using the validated French long version of the International Physical Activity questionnaire [25,26] (IPAQ). The IPAQ differentiates three levels of PA, from “slow” (level 1), “moderate” (level 2), to “vigorous” (level 3) according to the level of Metabolic equivalent of task (MET), a measure of energy expenditure. It investigates the time spent on intense activities, moderate activities, and walking in the past seven days [25].

Alcohol use and AUD severity were also assessed using the AUDIT [5,26], and the SADQ [4]. Concurrent nicotine and cannabis addictions were assessed using the Fagerström Test for Nicotine Dependence (FTND) [27] and the Cannabis Abuse Screening Test (CAST) [28]. Participants were asked if they had a concurrent opioid use and/or a concurrent stimulant use in the previous month. Depression symptoms, anxiety symptoms, and impulsivity features were assessed using the Beck Depression Inventory (BDI-II) [29], the State-Trait Anxiety Inventory (STAI-A and STAI-B) [30], and the UPPS Impulsive Behavior Scale (UPPS-Ps) [31], respectively.

Quality of sleep was scored using the Pittsburgh Sleep Quality Index (PSQI) [32] while the quality of life was evaluated using the World Health Organization Brief Quality of Life Assessment (WHOQOL-Bref) [33].

### 2.3. Statistical Analyses

Participants were divided into two groups according to their IPAQ score

Those with a “low” level of PA were classified in the “IPAQ level-1” group, whereas those with a moderate or vigorous level of PA were classified in the “IPAQ level-2&3” group. This recategorization was decided to facilitate the building of logistic regression models. Furthermore, we hypothesized that the “low” level should be the priority target of physical training interventions in AUD patients while “intermediate” and “high” levels of activity could be considered sufficient.

Bivariable comparisons of the two groups of interest were performed using the chi-squared test or Fisher’s exact test for categorical variables, and the Wilcoxon test for quantitative variables. We built multivariable logistic regression models, and adjusted for age, gender, marital status, level of education, and professional status. The dependent variable was the level of PA (IPAQ level-1 vs. IPAQ level-2&3). Explanatory variables were BMI (quantitative score), AUD DSM-5 category (severe versus not severe), AUD DSM-5 score (number of criteria), AUDIT score, SADQ score, FTND score, CAST score, concurrent opioid use (Yes/No) and/or a concurrent stimulant use (Yes/No), BDI-II score, STAI-A and STAI-B scores, UPPS-Ps score, WHOQOL-Bref subscores (environmental, social, physical, psychological) and PSQI score. Validity conditions were met to perform the logistic regressions. Subjects with missing values were not included in the analyses. All statistical analyses were performed using the XLSTAT software, Addinsoft (https://www.xlstat.com/en/, accessed on 25 February 2022).

## 3. Results

We included 382 participants in the analyses. A total of 103 participants were categorized as undertaking “insufficient physical activity” (IPAQ level-1 group), while 279 others were categorized as undertaking “sufficient physical activity” (IPAQ level-2&3 group). The median age of the sample was 38.0 [29.0–49.0] years, females represented 30.4% of the sample. The complete descriptive variables are displayed in Table 1.

The IPAQ level-1 group reported greater depression symptoms at the BDI-II (28.0, IQR [19.25;36.75] versus 17.0, IQR [11.0–27.0]) as well as higher degrees of anxiety at both the STAI-A (55.0, IQR [45.0;62.0] versus 47.0, IQR [36.0;58.75] and STAI-B (58.0, IQR [52.0;67.0] versus 55.0, IQR [46.0;63.0]). Scores for impulsivity were higher in the IPAQ level-1 group (54.0, IQR [47.0;61.5] versus 51.0, IQR [44.0;58.0]).

Participants in the IPAQ level-1 group reported higher dependence on nicotine (5.0, IQR [1.0;7.0] versus 3.0, IQR [0.0;6.0]) and a slightly higher prevalence of concurrent opioid use (18.6% versus 11.3%). There was no difference between the two groups regarding reported cannabis abuse or concurrent stimulant use. All subscores for quality of life were significantly lower in the IPAQ level-1 group, and participants of this group had a significantly deteriorated quality of sleep, scoring higher on the PSQI (Table 1).

When adjusting for age, gender, marital status, and level of education, the number of DSM-5 criteria met for AUD (aOR per each one-criterion increase: aOR:1.15, 95%CI [1.01;1.30]), and the concurrent opioid use (vs. no concurrent opioid use: aOR: 2.17, 95%CI [1.11;4.26]) were significant risk factors for classification into the IPAQ level-1 group. Similarly, increased scores (per one-point increase for each) at BDI-II (aOR 1.06, 95%CI [1.02;1.06]), STAI-A (aOR 1.04, 95%CI [1.02;1.06], STAI-B (aOR: 1.03, 95%CI [1.01;1.06], PSQI (aOR: 1.07, 95%CI [1.01;1.14]), UPPS-Ps (aOR: 1.04, 95%CI [1.01;1.14], and BMI (aOR: 1.08, 95%CI [1.03;1.15]) were associated with a higher risk of displaying a low level of PA. By contrast, greater WHOQOL-Bref scores in the environment (aOR: 0.98, 95%CI [0.96;0.99]), physical (aOR: 0.97, 95%CI [0.95;0.98]), psychological (aOR: 0.97, 95%CI [0.96;0.98]), and social domains (aOR: 0.98, 95%CI [0.97;0.99]) were protective against displaying an insufficient PA. Levels of significance are displayed in Table 2.

## 4. Discussion

This study aimed to explore the features associated with insufficient PA in patients with AUD. Overall, insufficient PA was associated with an increased number of DSM-5 criteria met for AUD, concurrent opioid use, greater levels of depression and anxiety, higher impulsivity, more pronounced sleep disorders, higher BMI, and lower scores in quality of life. To our knowledge, this is the first study that explored the features of insufficient PA in AUD. However, quality of life, quality of sleep, and BMI are also markers of global health in the general population [34,35,36]. The results of this study should thus be interpreted with caution, as the link found between PA and those factors could be independent of alcohol consumption.

As anxiety, impulsivity, depression symptoms, sleep disorders, concurrent opioid use and impaired quality of life are factors related to the severity of AUD, insufficient PA seems to reflect many aspects of the severity of AUD. Moreover, the more severe the AUD is, the more functional impairment it causes [37]. Greater severity of AUD could thus lead to a significant decrease in PA, up to physical deconditioning. For instance, it has been shown in previous studies that AUD patients demonstrate a significantly reduced work capacity, a reduced walking duration [38], impaired muscle strength [39], and a significant increase in heart rate at rest and during light-to-medium intensity exercise [40], compared to healthy subjects. Low PA and severe AUD could then mutually reinforce each other.

Most interventional studies exploring the effects of PA rehabilitation on AUD patients do not distinguish them according to their baseline level of PA [16,17,18]. In the studies of Hallgren et al. and Georgakouli et al., all the participants of the study were physically inactive at baseline [15,16,17] while PA level is used as a continuous measure for all participants in the study of Vancampfort et al. [18], with no particular categories of patients made regarding their level of PA. Our study emphasizes that targeting the subgroup of insufficient PA AUD patients for physical readaptation could be more accurate and more cost-efficient, as they present more functional impairments than AUD patients with sufficient PA. Similarly, no particular distinction is usually made regarding AUD patients with comorbid physical, psychiatric, or other substance use disorders [16,17] even though those subgroups could experience even greater benefits from physical rehabilitation programs. Furthermore, it is possible that these specific populations would experience greater difficulties adhering to this intervention due to their illness symptoms or medication side effects. For instance, it has been shown that those two factors, as well as many others, represent barriers to adopting and maintaining an active lifestyle in the population of adults suffering from mental illness [41]. These subgroups may present particular features and needs; further studies should thus be conducted to design more customized physical rehabilitation programs adapted to patients with comorbidities. For instance, it is possible that depressed AUD patients would need their depression to be treated prior to the initiation of the physical rehabilitation program, or that this program should be adapted to be more progressively added to the treatment sequence.

Another important question to be explored in the future is whether physical rehabilitation programs for AUD patients, particularly those with a low baseline level of PA, increase the level of PA in the long term, and whether AUD patients maintain the gain of the intervention as well as the improvements on their AUD symptoms after the end of the program. This is crucial, especially given that, in several studies, a significant part of the AUD patients without any specified physical, psychiatric or addictive comorbid disease dropped out of attending the PA sessions during the study [42,43].

This study has several limitations. First, there was a significant number of missing values for the quoting of AUD. Although clinicians always assessed the diagnosis of AUD using the eleven criteria of the DSM-5, and wrote down the diagnosis in the medical file of the patient, the supplemental separated file containing the detailed AUD DSM-5 criteria was, occasionally, not quoted properly. Because this information had not been accurately recorded during the medical consultation, details of which criteria of the DSM-5 were met for the diagnosis of AUD could not be extracted afterward for some patients. as clinicians assessed the diagnosis without systematically quoting each item in the clinical record. Second, as patients were self-reporting their level of PA, there could be reporting biases. However, the fact that the IPAQ assesses the level of PA in the past seven days reduces the probability of a memory bias. Third, we did adjust for age and level of education but we could not control for other potential confusion factors such as socio-economic factors that could mediate the relationship between AUD and PA. Last, this was a monocentric study and needs to be replicated in a multicentric study for generalization.

In conclusion, PA level is correlated with the severity of AUD. Targeting severe AUD patients for the implementation of PA rehabilitation in clinical settings could be more accurate and cost-efficient. Further studies are needed to design specific physical rehabilitation programs for AUD patients with physical, psychiatric, and addictive comorbidities.

## Figures and Tables

**Table 1 nutrients-14-04958-t001:** Bivariable comparisons of the IPAQ level-1 group and the IPAQ level-2&3 group.

Characteristics	Full Sample (*n* = 382)	IPAQ Level-1 ^1^ (*n* = 103)	IPAQ Level-2&3 ^2^ (*n* = 279)	*p*-Value	*n* Missing Values
Age (y), med [IQR] ^3^	38.0 [29.0–49.0]	41.0 [31.5–49.5]	38.0 [28.0–49.0]	0.09	0
Gender (females), *n* (%)	116 (30.4%)	37 (35.9%)	79 (28.3%)	0.15	0
Having a partner, *n* (%)	146 (38.3%)	37 (36.3%)	109 (39.1%)	0.65	1
Level of education (y after BD ^4^), med [IQR]	4.0 [3.0–6.0]	3.0 [3.0–5.0]	5.0 [3.0–6.0]	<0.01	4
Professional Status (active), *n* (%)	180 (47.1%)	37 (35.9%)	143 (50.9%)	<0.01	0
Body Mass Index (kg/m^2^), med [IQR]	23.7 [21.2–26.8]	24.8 [21.6–28.2]	23.3 [21.0–25.9]	<0.01	7
AUD ^5^ DSM-5 ^6^ category (*n* ‘severe’), *n* (%)	146 (64.0%)	48 (73.9%)	98 (62.4%)	0.10	160
AUD DSM-5 score (*n*), med [IQR]	7.0 [5.0–9.0]	8.0 [5.0–10.0]	6.0 [4.0–9.0]	<0.01	160
AUDIT ^7^ score (*n* of points), med [IQR]	23.0 [14.0–31.0]	26.0 [16.0–32.5]	22.0 [13.5–30.0]	0.03	0
SADQ ^8^ score (*n* of points), med [IQR]	11.0 [4.0–22.0]	16.0 [7.0–27.8]	10.0 [3.0–20.0]	<0.001	11
FTND ^9^ score (*n* of points), med [IQR]	4.0 [0.0–6.0]	5.0 [1.0–7.0]	3.0 [0.0–6.0]	<0.001	1
CAST ^10^ score (*n* of points), med [IQR]	0.0 [0.0–10.0]	0.0 [0.0–11.0]	0.0 [0.0–11.0]	0.36	8
Concurrent opioid use (*n*, %)	50 (13.3%)	19 (18.6%)	31 (11.3%)	0.06	5
Concurrent stimulant use (*n*, %)	126 (33.0%)	32 (31.1%)	94 (33.7%)	0.63	0
BDI-II ^11^ score (*n* of points), med [IQR]	20.0 [12.0–29.0]	28.0 [19.25–36.75]	17.0 [11.0–27.0]	<0.001	19
STAI-A ^12^ score (*n* of points), med [IQR]	50.0 [38.0–59.0]	55.0 [45.0–62.0]	47.0 [36.0–58.75]	<0.001	16
STAI-B score (*n* of points), med [IQR]	56.0 [48.0–65.0]	58.00 [52.0–67.0]	55.0 [46.0–63.0]	<0.001	17
UPPS-Ps ^13^ score (n of points), med [IQR]	52.0 [45.0–58.0]	54.0 [47.0–61.5]	51.0 [44.0–58.0]	0.03	3
PSQI ^14^ score (*n* of points), med [IQR]	9.0 [6.75–13.0]	11.0 [8.0–14.0]	9.0 [6.0–12.0]	<0.001	3
WHOQOL-Bref ^15^-env score (*n* of points), med [IQR]	63.0 [50.0–69.0]	56.0 [44.0–66.0]	63.0 [50.0–75.0]	<0.001	0
WHOQOL-Bref-soc score (*n* of points), med [IQR]	50.0 [31.0–69.0]	31.0 [19.0–56.0]	50.0 [31.0–69.0]	<0.001	0
WHOQOL-Bref-psy score (*n* of points), med [IQR]	44.0 [25.0–56.0]	31.0 [19.0–44.0]	44.0 [31.0–59.5]	<0.001	0
WHOQOL-Bref-phy score (*n* of points), med [IQR]	53.0 [38.0–69.0]	38.0 [31.0–56.0]	56.0 [44.0–69.0]	<0.001	0

^1^ Insufficient physical activity group, ^2^ sufficient physical activity group, ^3^ Interquartile range, ^4^ years of education after High school diploma, ^5^ Alcohol Use Disorder in the 5th Edition of the ^6^ Diagnostic and Statistical Manual of Mental Disorders, ^7^ Alcohol Use Disorder Identification Test, ^8^ Severity of Alcohol Dependence Questionnaire, ^9^ Fagerström Test for Nicotine Dependence, ^10^ Cannabis Abuse Screening Test, ^11^ Beck Depression Inventory, ^12^ State-Trait Anxiety Inventory, ^13^ Impulsivity scale, ^14^ Pittsburgh Sleep Quality Inventory, ^15^ WHO Quality of Life (environment, social, psychological, physicalCompared to individuals in the IPAQ level-2&3 group, those in the IPAQ level-1 group had a higher AUD DSM-5 score (8, IQR [5;10] versus 6, IQR [4;9]), higher AUDIT scores regarding alcohol consumption (26.0, IQR [16.0;32.5] versus 22.0, IQR [13.5;30.0]) and higher SADQ scores (16.0, IQR [7.0;27.8] versus 10.0, IQR [3.0;20.0]).

**Table 2 nutrients-14-04958-t002:** Multivariable logistic regression models, each adjusted for age, gender, marital status, level of education, and professional status.

	Bivariable Comparisons OR ^1^ (95%CI ^14^)	Adjusted Comparisons aOR ^2^ (95%CI)
Age	1.01 (1.00–1.03)	
Gender (ref: male)	1.42 (0.88–2.29)	
Marital Status (ref: partner)	0.89 (0.55–1.42)	
Level of education	0.83 (0.73–0.94) ***	
Professional Status (ref: active)	0.54 (0.34–0.86) *	
Body Mass Index	1.08 (1.03–1.14) ***	1.08 (1.03–1.15) ***
AUD ^3^ DSM-5 ^4^ category (ref: severe)	1.70 (0.90–3.23)	1.90 (0.96–3.43)
AUD DSM-5 score	1.16 (1.03–1.31) *	1.15 (1.01–1.30) *
AUDIT ^5^ score	1.03 (1.01–1.05) *	1.02 (0.99–1.04)
SADQ ^6^ score	1.03 (1.01–1.05) ***	1.02 (1.0–1.04) *
FTND ^7^ score	1.35 (1.19–1.47) **	1.15 (1.06–1.25) ***
CAST ^8^ score	1.02 (0.99–1.05)	1.03 (0.99–1.06)
Concurrent opioid use (ref: yes)	1.80 (0.97–3.36)	2.17 (1.11–4.26) *
Concurrent stimulant use (ref: yes)	0.89 (0.55–1.44)	0.96 (0.55–1.68)
BDI-II ^9^ score	1.06 (1.04–1.08) ***	1.06 (1.04–1.09) ***
STAI-A ^10^ score	1.03 (1.02–1.05) ***	1.04 (1.02–1.06) ***
STAI-B score	1.03 (1.01–1.06) ***	1.03 (1.01–1.06) ***
UPPS-Ps ^11^ score	1.01 (1.00–1.01)	1.04 (1.01–1.07) *
PSQI ^12^ score	1.09 (1.03–1.15) ***	1.07 (1.01–1.14) *
WHOQOL-Bref ^13^-env score	0.97 (0.96–0.99) ***	0.98 (0.96–0.99) ***
WHOQOL-Bref-soc score	0.98 (0.97–0.99) ***	0.98 (0.97–0.99) ***
WHOQOL-Bref-psy score	0.97 (0.96–0.98) ***	0.97 (0.96–0.98) ***
WHOQOL-Bref-phy score	0.96 (0.95–0.98) ***	0.97 (0.95–0.98) ***

^1^ Odds-ratio, ^2^ Adjusted Odds-ratio, ^3^ Alcohol Use Disorder in the 5th Edition of the ^4^ Diagnostic and Statistical Manual of Mental Disorders, ^5^ Alcohol Use Disorder Identification Test, ^6^ Severity of Alcohol Dependence Questionnaire, ^7^ Fagerström Test for Nicotine Dependence, ^8^ Cannabis Abuse Screening Test, ^9^ Beck Depression Inventory, ^10^ State-Trait Anxiety Inventory, ^11^ Impulsivity scale, ^12^ Pittsburgh Sleep Quality Inventory, ^13^ WHO Quality of Life (environment, social, psychological, physical), ^14^ Confidence Interval. No. * *p* < 0.05, ** *p* < 0.01, *** *p* < 0.001.

## Data Availability

No application.

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
