# Peer review of "Insufficient Physical Activity Is a Global Marker of Severity in Alcohol Use Disorder: Results from a Cross-Sectional Study in 382 Treatment-Seeking Patients"

_nutrients, 2022, doi:10.3390/nu14234958_

Round 1
Reviewer 1 Report
nutrients-2021470_review
Title: Insufficient physical activity is a global marker of severity in alcohol use disorder: results from a cross-sectional study in 382 treatment-seeking patients
Comments and Suggestions for Authors
Dear authors,
I have carefully read your paper, which investigated differences in physical activity levels in patients with alcohol use disorder and its association with common markers of severity of the illness.
Your results show that insufficient physical activity appears to be associated with a greater number of met DSM-5 criteria for alcohol use disorders, concurrent use of opioids, higher levels of depression and anxiety, greater impulsivity, more pronounced sleep disturbances, higher BMI and lower quality of life scores. Therefore, it appears that severe levels of alcohol use disorders correlate with low levels of physical activity.
In general, the manuscript is well-written. The text is understandable and organized, and it is easy to follow authors’ thoughts and reasoning. However, I found some issues in introduction, materials and methods section, analyses and conclusions that should be addressed to improve the paper, in my opinion.
Specific comments:
Introduction
I suggest you to add some information regarding the prevalence of alcohol use disorders globally and in your country or region. This can help to contextualize the importance of this disease and the need to carry out studies on this topic.
Material and methods
- Page 2, lines 64-71. Please, could you add more information about where the data was recorded? How long did the patients take to complete all the questionnaires? Was someone supervising the patients while they were taking the questionnaires?
- Page 2: What inclusion or exclusion criteria were established to select the participants in the study? Please add this information
- Page 2 lines 64-71: You mentioned that participants had been diagnosed with AUD during their first medical consultation, using the DSM-5 criteria, however, you recognize as a limitation of your study that there was a significant number of missing values for the quoting of AUD as clinicians assessed diagnosis without systematically quoting each item in the clinical record.
I therefore understand that it is necessary to analyze the 11 items established by the DSM 5 to make the diagnosis AUD. I wonder then, how the doctors did not record this information in detail or why you could not access this information that is crucial for your study.
- Page 3. You point out that this study was conducted on a large sample of AUD patients seeking treatment. However, there is no statement about the sample size. How were the sample size calculated? This information is very important to your study. Please add this information.
- Page 2, lines 73-94. If possible, report the reliability and validity of the different scales and questionnaires that you mentioned in the result section and add some references that supports it, if necessary.
- Page 3, lines 95-115. The statistical analysis is correct; however, in my opinion, it could have been more complete if you had left the classification of physical activity in the three groups established by The International Physical Activity questionnaire. IPAQ level-1“low”; IPAQ level-2“moderate”; IPAQ level-3“vigorous”
In this way, you could have made more possible combinations ( Level 1 vs level 2; level 1 vs level 3; level 2 vs level 3) and obtain broader results.
Results
The results section is well-structured and comprehensive. As I mentioned in my previous comment, your results could have been more complete. Also, I suggest performing a flow diagram to facilitate monitoring the entire study participants.
Discussion
Your discussion section is in general adequate.
As yourselves acknowledge, the results of your study should thus be interpreted with caution, so I suggest you to the reformulate your conclusions in a more carefully way.
I hope that my comments could help to improve the paper.
Congratulations for your research.
Author Response
Introduction
I suggest you to add some information regarding the prevalence of alcohol use disorders globally and in your country or region. This can help to contextualize the importance of this disease and the need to carry out studies on this topic.
We kindly thank the reviewer for this suggestion, and we have added information regarding international and national prevalence of alcohol use disorder at the beginning of our introduction.
“According to the last Global status report on alcohol and health from the World Health Organization, in 2016, 2.3 billion people were current alcohol drinkers worldwide, and 5.1% of the adults older than 15 years old suffered from an Alcohol Use Disorder (AUD) [1]. Harmful use of alcohol was responsible for 3 million deaths (representing 5.3% of all deaths) [1]. In France, 10% of the adults were using alcohol daily in 2020, and alcohol was responsible for 41,000 annual deaths [2]. The French national prevalence of AUD is estimated to be 7.0% [1].”
Material and methods
Page 2, lines 64-71. Please, could you add more information about where the data was recorded? How long did the patients take to complete all the questionnaires? Was someone supervising the patients while they were taking the questionnaires?
We agree with the reviewer that this part of the Materials and Methods section needed some additional details, which we have provided in the appropriate section.
“The questionnaires were filled by participants within the university addiction consultation center and the answered were coded and stored by one of the investigators in an electronic database. It generally took one hour for the participants to fully complete the questionnaires. The investigator in charge of collecting the data was supervising the participants while they were filling the questionnaires, to help them clarify the meaning of the questions when it was needed.”
Page 2: What inclusion or exclusion criteria were established to select the participants in the study? Please add this information
We assume that the details provided regarding inclusion and exclusion criteria may not have been highlighted clearly enough in our manuscript. We have modified our manuscript so that inclusion and exclusion criteria appear more easily for the reader.
“Included participants were adult, aged 18 years old or more, male and female outpatients of a French expert university addiction consultation center who had been diagnosed with AUD during their first medical consultation, using the DSM-5 criteria. Participants not fulfilling one or more of those inclusion criteria were excluded.”
Page 2 lines 64-71: You mentioned that participants had been diagnosed with AUD during their first medical consultation, using the DSM-5 criteria, however, you recognize as a limitation of your study that there was a significant number of missing values for the quoting of AUD as clinicians assessed diagnosis without systematically quoting each item in the clinical record.
I therefore understand that it is necessary to analyze the 11 items established by the DSM 5 to make the diagnosis AUD. I wonder then, how the doctors did not record this information in detail or why you could not access this information that is crucial for your study.
We agree with the reviewer that this point can seem illogical. Clinicians were always assessing the diagnosis of AUD using the DSM-5 eleven criteria, and writing the final diagnosis in the patient file. However, they were not always quoting electronically the detached document provided with the detailed 11 DSM-5 criteria for AUD. Therefore, details of which criteria of the DSM-5 was met for AUD could not always be extracted afterwards for some patients, because the information had not been properly recorded at the time of the consultation. We have modified our manuscript and hope that there is no more misinterpretation of this information.
“Although clinicians always assessed the diagnosis of AUD using the eleven criteria of the DSM-5, and wrote down the diagnosis in the medical file of the patient, the supplemental separated file containing the detailed AUD DSM-5 criteria was, occasionally, not quoted properly. Because this information had not been accurately recorded during the medical consultation, details of which criteria of the DSM-5 was met for the diagnosis of AUD could not be extracted afterwards for some patients.”
Page 3. You point out that this study was conducted on a large sample of AUD patients seeking treatment. However, there is no statement about the sample size. How were the sample size calculated? This information is very important to your study. Please add this information.
We understand the legitimate questioning of the reviewer about that matter. No sample size was calculated for this study. We agree that calculation of the sample size is usually calculated in clinical studies. However, unlike randomized clinical trials, our study is exploratory and addresses objectives rather than specific hypothesis. Therefore, we believe calculating a sample size was not crucial for our design.
Page 2, lines 73-94. If possible, report the reliability and validity of the different scales and questionnaires that you mentioned in the result section and add some references that supports it, if necessary.
We thank the reviewer for this kind advice, but fear that detailing the indexes for reliability and validity for each of the scales we used will make this section more complex and difficult to read. Furthermore, it would provide the reader with details that have few links with the objectives of our study. However, in the Methods section, we have provided for each scale one or more studies assessing its validity, studies in which the reader will be able to find the detailed scores for reliability and validity.
Page 3, lines 95-115. The statistical analysis is correct; however, in my opinion, it could have been more complete if you had left the classification of physical activity in the three groups established by The International Physical Activity questionnaire. IPAQ level-1“low”; IPAQ level-2“moderate”; IPAQ level-3“vigorous”In this way, you could have made more possible combinations ( Level 1 vs level 2; level 1 vs level 3; level 2 vs level 3) and obtain broader results.
We thank the reviewer for his/her thoughtful remark and understand his/her point. We decided to use a classification with two levels instead of three because we were mainly interested in comparing the participants with a low physical activity (IPAQ-1 group) to the others (IPAQ-2 and IPAQ-3 groups), to assess whether physical activity rehabilitation in AUD patients should focus precisely on this specific population rather than on the two others. As such, we believe that a logistic regression with two categories was more fitted for this question than a multinomial regression.
Results
The results section is well-structured and comprehensive. As I mentioned in my previous comment, your results could have been more complete. Also, I suggest performing a flow diagram to facilitate monitoring the entire study participants.
We kindly thank the reviewer for his/her suggestions. We hope that the justification we have provided to his/her previous comment clarify our choice regarding the statistical analyses.
Although we take into account the second suggestion of the reviewer, our study has a cross-sectional design, thus, we believe there may not be a great gain of information in providing a flow diagram.
Discussion
Your discussion section is in general adequate.
As yourselves acknowledge, the results of your study should thus be interpreted with caution, so I suggest you to the reformulate your conclusions in a more carefully way.
As kindly suggested by the reviewer, we have slightly modified our conclusion section so that it is formulated in a more careful way.
“In conclusion, PA level is correlated with the severity of AUD. Targeting severe AUD patients for the implementation of PA rehabilitation in clinical settings could be more accurate and cost-efficient.”
I hope that my comments could help to improve the paper.
Congratulations for your research.
We thank the reviewer deeply for the time he/she has given to our manuscript, and for his/her many thoughtful suggestions that have helped us improve the quality of our manuscript.
Reviewer 2 Report
Methodological Niases exist
(The Authors must see my remarks)

Author Response
Please clarify the type of the article, eg. Research?
We thank the reviewer for his/her carefulness and have added the requested information (Research article).
Was the questionnaire standardized? If so, state Reference...
We understand that the reviewer is referring to the questions pertaining to the sociodemographic dimensions. We did not use a standardized questionnaire for these questions, as age, gender, marital status, level of education, professional status and body mass index are variables usually used as they are in epidemiological studies.
How did the Authors determine the study sample? Inclusion/Exclusion Criteria? Protocol? Reference(s)?
We thank the reviewer for giving us the opportunity to clarify our methodology. We assume that the details provided regarding inclusion and exclusion criteria may not have been highlighted clearly enough in our manuscript. We have modified our manuscript so that inclusion and exclusion criteria appear more easily for the reader.
“Included participants were adult, aged 18 years old or more, male and female outpatients of a French expert university addiction consultation center who had been diagnosed with AUD during their first medical consultation, using the DSM-5 criteria. Participants not fulfilling one or more of those inclusion criteria were excluded.”
No sample size was calculated for this study. Sample size is usually calculated in clinical studies but, unlike randomized clinical trials, our study is exploratory and addresses objectives rather than specific hypothesis. Therefore, we believe calculating a sample size was not as crucial for our design.
Do not state personal opinions or Conclusions....
As advised by the reviewer, we have modified this part of the discussion so that there is no more feeling that we state our personal opinions, but rather that we consider this statement about depressed AUD patients as being a possibility or hypothesis that should be further explored.
“For instance, it is possible that depressed AUD patients would need their depression to be treated prior to the initiation of the physical rehabilitation program, or that this program should be adapted to be more progressively added to the treatment sequence.”
State a special Section ''Conclusion(s)
We thank the reviewer for his/her kind suggestion. We believe we have clearly identified the conclusion section and assume there may not be any further need to state this section as the ‘Conclusion’ section, unless specified otherwise by the editors.
We thank the reviewer deeply for having helped us improve the quality of our manuscript. We have added adequate references from the scientific literature according to what he/she requested.
Reviewer 3 Report
The subject of the work is the role of the level of physical activity in patients with alcohol use disorders.
To the authors.
Your job is interesting, it must have required a lot of effort.
Have you submitted your research as a clinical trial study?
Unfortunately, the presentation of the results in the results section is not entirely clear, especially in the description part.
Can you complete the group-specific average duration of alcohol use disorder?
Author Response
The subject of the work is the role of the level of physical activity in patients with alcohol use disorders.
To the authors.
Your job is interesting; it must have required a lot of effort.
Have you submitted your research as a clinical trial study?
We thank the reviewer for his/her kind compliments, we have submitted our observational study as a research article.
Unfortunately, the presentation of the results in the results section is not entirely clear, especially in the description part.
We thank the reviewer for his/her comment, but don’t fully understand how he/she thinks this part could be improved.
Can you complete the group-specific average duration of alcohol use disorder?
We agree with the reviewer that this information could have been relevant, but we could not extract it during the study, as this specific question was not asked in our questionnaire. We considered assessing the severity of the alcohol use disorder by using the DSM-5 criteria provided enough information for our research question.